# Phylogeny and Taxonomy of the Genus *Amphichorda* (*Bionectriaceae*): An Update on Beauveria-like Strains and Description of a Novel Species from Marine Sediments

Daniel Guerra-Mateo [1], Josepa Gené [1,2,*], Vladimir Baulin [2,3] and José Francisco Cano-Lira [1,2]



1   Unitat de Micologia i Microbiologia Ambiental, Facultat de Medicina i Ciències de la Salut, Universitat Rovira i Virgili, 43201 Reus, Spain; daniel.guerra@urv.cat (D.G.-M.); jose.cano@urv.cat (J.F.C.-L.)
2   Institut Universitari de Recerca en Sostenibilitat, Canvi Climàtic i Transició Energètica (IU-RESCAT), Universitat Rovira i Virgili, 43007 Tarragona, Spain; vladimir.baulin@urv.cat
3   Física i Cristal·Lografia de Materials, Escola Tècnica Superior d'Enginyeria Química, Universitat Rovira i Virgili, 43007 Tarragona, Spain
*   Correspondence: josepa.gene@urv.cat; Tel.: +34-977759359

**Abstract:** The genus *Amphichorda* has been recently re-erected as an independent lineage from *Beauveria*, circumscribed within *Cordycipitaceae*. However, its phylogenetic relationships with other members of this family remain obscure. In our ongoing efforts to expand the knowledge on the diversity of culturable ascomycetes from the Mediterranean Sea, we isolated several specimens of *Amphichorda*. Preliminary sequence analyses revealed great phylogenetic distance with accepted *Amphichorda* species and a close relation to *Onychophora coprophila*. *Onychophora* is a monotypic genus of enteroblastic conidiogenous cells, presumably related to *Acremonium* (*Bionectriaceae*), while *Amphichorda* exhibits holoblastic conidiogenesis. Here, we examine representative strains of *Amphichorda* species to elucidate the taxonomy of the genus and the aforementioned fungi, combining morphological, ultrastructure and multi-locus phylogenetic analyses (ITS, LSU, *tef*1, *BenA*). The results revealed *Amphichorda* as a member of the *Bionectriaceae*, where its asexual morphs represent a transition between enteroblastic and holoblastic conidiogenesis for this group of fungi. We also describe and illustrate *Amphichorda littoralis* sp. nov. and propose *Amphichorda coprophila* comb. nov. In addition, we establish key phenotypic features to distinguish *Amphichorda* species and demonstrate the higher salt tolerance degree of *A. littoralis*, consistent with its marine origin. This work provides a comprehensive framework for future studies in the genus.

**Keywords:** *Ascomycota*; asexual fungi; marine fungi; multi-locus phylogeny; new taxa; ultrastructure

## 1. Introduction

The order *Hypocreales* (*Sordariomycetes*, *Pezizomycotina*, *Ascomycota*) currently comprises around 300 genera distributed across 17 families. Their species inhabit a wide range of substrates in terrestrial and aquatic (marine and freshwater) environments, and they show a great variety of lifestyles, such as saprobic, endophytic, and pathogenic fungi in plants and animals, including humans [1,2]. *Cordycipitaceae* are one of the most complex families in the order due to the pathogenic behavior of most of its species, which include a wide range of invertebrate hosts, and this results in a variety of morphological features of the sexual morph, primarily associated with its ascomata (stroma and perithecia). The asexual morphs, on the contrary, are very similar, most frequently showing phialidic conidiogenous cells. Therefore, genera such as *Amphichorda*, *Beauveria*, *Cordyceps* or *Isaria* have been difficult to delineate, and recent works have dealt with numerous taxonomical problems [3,4]. In particular, the genus *Amphichorda* has been traditionally accepted as a member of *Cordycipitaceae* based on the taxonomical history of its type species, *Amphichorda felina* (=*Beauveria felina*). Despite this, the most recent multi-locus phylogeny of the family

has resolved *Amphichorda* as a sister lineage to the main *Cordycipitaceae* clade [4]. Indeed, the taxonomic status of *Amphichorda* has been controversial since its original description.

The genus *Amphichorda* was described by Fries in 1825 and typified by *A. felina*, a fungus isolated from cat dung in France and previously classified in the genus *Clavaria* [5,6]. The morphological description stated the white farinaceous color of the colonies and the production of filiform conidiogenous cells, which inspired the name of the genus. In 1832, *A. felina* was transferred to the genus *Isaria*, a genus that, at that time, lacked a type species and, therefore, comprised morphologically heterogenous fungi [7,8]. In 1972, de Hoog redefined *Isaria* based on the production of synnemata and, following von Arx, accepted *Isaria felina* as the lectotype of the genus [9]. This circumscription, however, was rejected due to the previous lectotypification with the species *Isaria farinosa* [8,10]. *Isaria felina* was then transferred to the genus *Beauveria* based on the morphological resemblance of their holoblastic conidiogenous cells. However, early phylogenetic analyses suggested great dissimilarity between *B. felina* and other *Beauveria* species [8]. Recently, the genus *Amphichorda* has been re-erected for the description of two novel species, *Amphichorda cavernicola* and *Amphichorda guana*, by Zhang et al. [11,12]. Zhang's studies represent the first phylogenetic backbone for the genus *Amphichorda* and demonstrate the great phylogenetic distance between *Amphichorda* and *Beauveria*. Despite this, the phylogenetic relationships between *Amphichorda* and other genera remain obscure. Moreover, the type material of *A. felina* seems to be lost. Zhang et al. considered the strain CBS 250.34, the type of *Isaria cretaceae*, as the type strain of *A. felina* [12]. *Isaria cretaceae* was synonymized with *A. felina* by de Hoog [9]; however, so far, this strain has not been designated as the epitype of the species. Nevertheless, the type strain of *I. cretacea* was isolated from a package of moldy pressed yeast from Epsom, England [13], and, according to the criteria of substrate and geographical similarity required for fungal epitypification [14], the strain CBS 250.34 could not represent *A. felina* on the basis of its coprophilous origin from France [5,6]. Thus, the taxonomical status and the phylogenetic relationships of *Amphichorda* need to be revised.

During a survey of culturable ascomycetes from the Mediterranean Sea, we isolated several interesting specimens of an amphichorda-like fungus. A preliminary sequence analysis of the nuclear ribosomal region (i.e., the 28S large ribosomal subunit (LSU) and the internal transcribed spacer (ITS), including the 5.8S rDNA gene) showed that these specimens belong to the genus *Amphichorda*, but they do not fit into any of the described species. This preliminary sequence analysis also revealed that the marine strains are closely related to *Onychophora coprophila*. *Onychophora* is a monotypic genus, and the conidiogenous apparatus morphologically resembles *Amphichorda* species. However, the conidiogenous cells of the former were described as enteroblastic (phialidic), while *Amphichorda* exhibits holoblastic conidiogenesis [11,12,15]. In addition, according to the MycoBank and Index Fungorum databases, *O. coprophila* is a fungus of uncertain position among *Ascomycota*, although the original authors suggested a possible relation to *Acremonium* (*Bionectriaceae*) [15].

The aim of this study was to clarify the taxonomy of the above-mentioned fungi based on morphological features, including the ultrastructure, and multi-locus phylogenetic analyses inferred with sequences of the nuclear markers available for *Amphichorda* species. These are the ITS and LSU regions of the rDNA and partial fragments of the translation elongation factor 1-α (*tef*1) and the β-tubulin (*BenA*) regions. We used these nuclear markers to determine the strains phylogenetically related to the fungi under study and to examine their available living cultures in order to assess the diversity within *Amphichorda*. In this work, we provide an update on the morphological and molecular diversity of *Amphichorda*, determine its phylogenetic relationships with other genera in *Hypocreales* and discuss the type strain for *A. felina*.

## 2. Materials and Methods

### 2.1. Sampling and Strains

Sediment samples were collected from the Mediterranean Sea at two points of the Tarragona coast, the Miracle and Arrabassada beaches, through 2021 and 2022. Each beach

was sampled twice, the Miracle beach in June and October 2021 and the Arrabassada beach in February and June 2022. These beaches are located in the southern part of Catalonia, right next to the port of Tarragona, the fifth most important harbor in Spain and an important stop for tourism cruise ships [16].

Marine sediments were collected following the same methodology at both locations. We established four collection points based on the sediment grain size and depth in the column of water. The first point was at 6 m depth (sand sediment), the second at 13 m (sand sediment), the third at 20 m (transition between sand and silt sediment) and the last point was at 30 m depth (silt sediment). Four sub-samples were collected at each point ca. 15 cm below the surface of the sea bed using 50 mL sterile plastic containers. They were transported in a portable refrigerator to the laboratory and processed immediately. For each sampling point, sediment sub-samples were mixed and vigorously shaken in a container; after resting for 1 min, the water was decanted, and the sediment was deposited into plastic trays on sterile filter paper to remove excess water.

Three agar media were used to achieve a greater ascomycetous diversity in culture and restrict the growth of certain fungal groups: dichloran rose-bengal chloramphenicol agar (DRBC; 5 g peptone, 10 g glucose, 1 g $KH_2PO_4$, 0.5 g $MgSO_4$, 25 mg rose-bengal, 200 mg chloramphenicol, 2 mg dichloran, 15 g agar, 1 L distilled water); 3% malt extract agar supplemented with seawater (SWMEA3%; 30 g malt extract, 5 g mycological peptone, 15 g agar, 1 L seawater), which was used as a suitable medium for the isolation of marine fungi [17]; and potato dextrose agar (PDA; Condalab, Madrid, Spain) supplemented with 2 g/L of cycloheximide (PDA+A), which was used to isolate strains resistant to this protein synthesis inhibitor, a frequent trait among fungi [18]. Both SWMEA3% and PDA+A culture media were supplemented with 5 mL of chloramphenicol (15 g/L ethanol) to prevent bacterial growth.

The culture methodology was as follows: 1 g of sediment from each sampling point was distributed across two Petri dishes and mixed with melted SWMEA3% at 45 °C; the same procedure was used to mix the sediment with PDA+A. In the case of the sediment mixed with DRBC, only 0.5 g of sediment from each sampling point was distributed across two Petri dishes to deal with fast-growing fungi. A set of the plates of the different culture media was incubated at 22–24 °C and the other set at 15 °C to enable the detection of slow-growing fungi. Plates were stored in darkness and examined every week under the stereomicroscope for 5–8 weeks. Pure cultures were obtained from tiny fragments of the colonies or conidia of the fungi from primary cultures using a sterile dissection needle and were cultured on PDA and incubated at 25 °C in the dark. These PDA cultures were used for preliminary morphological identification before DNA extraction.

Strains of potential novel or rare fungi were preserved and deposited in the culture collection of the Faculty of Medicine in Reus (FMR, Reus, Spain) for further studies. Taxonomic information and nomenclatural novelties were deposited in MycoBank. Cultures from the type strains and holotypes (i.e., dry colonies on the most appropriate media for their sporulation) were also deposited at the Westerdijk Fungal Biodiversity Institute in Utrecht (CBS, Utrecht, The Netherlands).

In addition to the strains from marine sediments, we revived another amphichordalike fungus from our fungal culture collection (FMR 17952). This strain was isolated from a fragment of a rubber tire floating in the seawater of the Miracle beach in July 2020. The tire fragment was washed three times with 10% NaClO (bleach) for 30 s, cut into small pieces and cultured on DRBC at 25 °C in darkness. We also examined several strains of *A. felina*, available in the CBS culture collection and labelled as *B. felina* (i.e., CBS 110.08, CBS 250.34, CBS 312.50, CBS 648.66 and CBS 173.71), to study the morphological and molecular variability of this species and for epitypification purposes based on the lack of a type strain for this species. Furthermore, the ex-type strain and a reference strain of *O. coprophila* (i.e., CBS 247.82 and CBS 424.88) were also added to the study due to the above-mentioned reasons (Table 1).

**Table 1.** GenBank accessions of the *Amphichorda* strains included in the present study.

| Species | Strain Number | Substrate (Country) | GenBank Accession Numbers [1] | | | | Citation |
|---|---|---|---|---|---|---|---|
| | | | ITS | LSU | *tef*1 | *BenA* | |
| *A. cavernicola* | CGMCC3.19571 [T] | Bird feces (China) | MK329056 | MK328961 | MK335997 | NA | [12] |
| | LC 12560 | Animal feces (China) | MK329061 | MK328966 | MK336002 | NA | [12] |
| | LC 12674 | Plant debris (China) | MK329065 | MK328970 | MK336006 | NA | [12] |
| *A. coprophila* | CBS 247.82 [T] (ex-type of *O. coprophila*) | Rabbit dung (England) | MH861494 | MH873238 | **OQ954487** | **OQ981138** | [19]; this study |
| | CBS 424.88 (received as *O. coprophila*) | Chipmunk dung (Canada) | **OQ942929** | **OQ943166** | **OQ954488** | **OQ981139** | This study |
| | CBS 173.71 (received as *B. felina*) | Porcupine dung (Canada) | AY261368 | MH871833 | **OQ954489** | **OQ981140** | [19]; this study |
| *A. felina* | CBS 250.34 (ex-type of *I. cretacea*) | Pressed yeast (England) | MH855498 | **OQ943167** | **OQ954490** | **OQ981141** | [19]; this study |
| | CBS 648.66 (received as *B. felina*) | Unknown (Argentina) | **OQ942930** | MH870575 | **OQ954491** | **OQ981142** | [19]; this study |
| | CBS 110.08 (received as *B. felina*) | Unknown | MH854578 | **OQ943168** | **OQ954492** | **OQ981143** | [19]; this study |
| *A. guana* | CGMCC3.17908 [T] | Bat guano (China) | KU746665 | KU746711 | KX855211 | NA | [11] |
| | CGMCC3.17909 | Bat guano (China) | KU746666 | KU746712 | KX855212 | NA | [11] |
| | CBS 312.50 (received as *B. felina*) | Rabbit dung (Unknown) | MH856641 | MH868150 | **OQ954493** | **OQ981144** | [19]; this study |
| *A. littoralis* | FMR 17952 | Floating rubber tire (Spain) | **OQ942925** | **OQ943162** | **OQ954483** | **OQ981134** | This study |
| | FMR 19404 [T] | Marine sediment (Spain) | **OQ942924** | **OQ943161** | **OQ954482** | **OQ981133** | This study |
| | FMR 19611 | Marine sediment (Spain) | **OQ942926** | **OQ943163** | **OQ954484** | **OQ981135** | This study |
| | FMR 20067 | Marine sediment (Spain) | **OQ942927** | **OQ943164** | **OQ954485** | **OQ981136** | This study |
| | FMR 20149 | Marine sediment (Spain) | **OQ942928** | **OQ943165** | **OQ954486** | **OQ981137** | This study |

CBS: culture collection of the Westerdijk Fungal Biodiversity Institute, Utrecht, The Netherlands; CGMCC: China General Microbiological Culture Collection Center, China; FMR: Facultat de Medicina i Ciències de la Salut, Reus, Spain. [T] indicates ex-type strains. "NA" indicates sequences not used in this study. [1] ITS: internal transcribed spacer region of the rDNA and 5.8S gene; LSU: 28S large ribosomal subunit; *tef*1: translation elongation factor 1α; *BenA*: tubulin. Sequences generated in this study are highlighted in bold.

### 2.2. Phenotypic Analysis

Microscopic features were obtained from strains growing on oatmeal agar (OA; 30 g oatmeal, 15 g agar, 1 L distilled water) after 14 days at 25 °C in darkness. Microscopic slides were mounted with lactic acid and observed with an Olympus BH-2 bright-field microscope (Olympus Corporation, Tokyo, Japan). In species descriptions, size ranges of relevant structures were derived from at least 30 measurements. Photomicrographs were obtained using a Zeiss Axio-Imager M1 light microscope (Zeiss, Oberkochen, Germany) with a DeltaPix Infinity digital camera. Due to the difficulties in observing the conidiogenic patterns in *Amphichorda* and *Onychophora* under a light microscope, representative strains of species of both genera were examined by scanning electron microscopy (SEM) using the Quanta 600 FEG Scanning Electron Microscope (Thermo Fisher Scientific, Waltham, MA, USA). The specimens were processed in accordance with Figueras and Guarro [20].

Macroscopic characterization of the colonies was made on PDA, OA and synthetic nutrient-poor agar (SNA; 1 g KH$_2$PO$_4$, 1 g KNO$_3$, 0.5 g MgSO$_{47}$·H2O, 0.5 g KCl, 0.2 g glucose, 0.2 g sucrose, 15 g agar, 1 L distilled water) after 14 days at 25 °C in darkness. Color notations in descriptions followed Kornerup and Wanscher [21]. Photoplates were assembled using GIMP v. 2.10.34 (GNU Image Manipulation Program).

In addition, we assessed the ability of *Amphichorda* species to grow at different temperatures by culturing the strains on PDA from 5 to 40 °C at intervals of 5 °C. Colony diameter was measured after 14 days in darkness. Moreover, the marine strains were com-pared with their phylogenetically related taxa to test a possible adaptation to the marine environment (salt tolerance). Cultures were carried out on solid agar plates of malt extract agar (MEA; 20 g malt extract, 15 g agar, 1 L distilled water) and MEA supplemented with 3.5% (35 ppt, the salt concentration in the marine environment), 5%, 10% and 15% NaCl. We used the strain *Aspergillus chevalieri* FMR 19829 (obtained from the marine sediments of this study) as a positive control with the ability to grow at high NaCl concentrations. Colony diameter was measured after 14 days and after 28 days at 25 °C. All tests were performed in duplicate, and the results represent the mean of the colony diameter between duplicate plates.

### 2.3. DNA Extraction, PCR Amplification and Sequencing

Total genomic DNA was extracted through the modified protocol of Müller et al. [22] and quantified using a Nanodrop 2000 (Thermo Scientific, Madrid, Spain). Four loci were used and amplified with the following primer pairs: ITS and LSU regions of the nrDNA with ITS5/LR5 [23,24], partial fragments of the *BenA* gene with T10/Bt2b [25] and the *tef*1 gene with EF-983F/EF-2218R [26]. Briefly, PCR conditions for ITS, LSU, *BenA* and *tef*1 were set as follows: an initial denaturation at 95 °C for 5 min, followed by 35 cycles of 30 s at 95 °C, 45 s at 56 °C and 1 min at 72 °C and a final extension step at 72 °C for 10 min. PCR products were purified and sequenced at Macrogen Corp. Europe (Madrid, Spain) with the same primers used for amplification. Consensus sequences were assembled using SeqMan v. 7.0.0 (DNAStar Lasergene, Madison, WI, USA).

The preliminary identification of the strains was performed by comparing their ITS regions with those available at the National Center for Biotechnology Information (NCBI) using the Basic Local Alignment Search Tool (BLAST; https://blast.ncbi.nlm.nih.gov/blast.cgi, accessed on 6 October 2022). Since intraspecific variation has been little studied for the ITS region in *Amphichorda*, a maximum similarity level of >98% with ≥90% of sequence coverage was used for species-level identification. Lower similarity values were considered as potential unknown fungi, and their taxonomic position was assessed with the aforementioned loci.

The sequences used for species-level identification were obtained from GenBank. In Table 1, the strains of *Amphichorda* and *Onychophora* and their GenBank accession numbers are listed. In Table S1 (Supplementary Material), information about representative strains of *Bionectriaceae*, *Cordycipitaceae* and outgroups used in the phylogenetic analyses is included.

### 2.4. Phylogenetic Analyses

The phylogenetic relationships of the strains examined in the present study were assessed using the ITS, LSU and *tef*1 regions. Sequences from each gene region were aligned independently in MEGA (Molecular Evolutionary Genetics Analysis) software v. 6.0 [27] using the ClustalW algorithm [28] and, when necessary, refined with MUS-CLE [29] or adjusted manually. Before combining the regions, the phylogenetic concordance between each individual phylogeny was tested through visual comparison to assess incongruent results among clades with high statistical support. When the concordance was confirmed, separate alignments were concatenated into a single data matrix in MEGA [27]. The partial fragments of the *BenA* gene were excluded from the phylogenetic analyses due to the limited availability of this region within the *Bionectriaceae* and, most notably, on taxa phylogenetically close to *Amphichorda*. However, *BenA* sequences were used to assess the similarity between *Amphichorda* species.

Maximum-likelihood (ML) analysis and Bayesian analysis (BA) were used for phylogenetic inference of individual sequence alignments and the concatenated alignment (ITS–LSU and ITS–LSU–*tef*1). ML analyses were conducted using the CIPRES Science Gateway portal v. 3.3 (https://www.phylo.org/, accessed on 5 December 2022; [30]) and RAxML-HPC2 on XSEDE v. 8.2.12 [31] with the default GTR substitution matrix and 1000 rapid bootstrap replications. Additional ML analyses were performed using IQ-TREE v. 2.1.2 [32,33] with ultrafast bootstrapping for the estimation of branch support [34]. The most suitable evolutionary model for each partition was estimated using ModelFinder [35,36], implemented in IQ-TREE. Bootstrap support (bs) ≥70 was considered significant [37]. Bayesian analyses were performed using MrBayes v. 3.2.6 [38]. The best substitution model for each locus was estimated using jModelTest v. 2.1.3 following the Akaike criterion [39,40]. Markov chain Monte Carlo sampling (MCMC) was performed for 10 million generations using four simultaneous chains (one cold chain and three heated chains) starting from a random tree topology. Trees were sampled every 1000th generation or until the run was stopped automatically when the average standard deviation of split frequencies fell below 0.01. The first 25% of the trees was discarded as the burn-in phase of each analysis, and the remaining trees were used to calculate posterior probabilities (pp). A pp value of ≥0.95 was considered significant [41]. The resulting trees were plotted using FigTree v. 1.3.1 (http://tree.bio.ed.ac.uk/software/figtree/, accessed on 5 December 2022). The DNA sequences generated in this study were deposited in GenBank (Table 1), and the alignments were submitted to Zenodo (https://doi.org/10.5281/zenodo.7937438, accessed on 20 May 2023).

## 3. Results

Among the fungi detected from marine sediments collected at different depths, we recovered four strains (FMR 19404, FMR 19611, FMR 20067 and FMR 20149) exclusively from samples collected at 20 m depth at both the Miracle and Arrabassada beaches using SWMEA3% and DRBC culture media. These strains and FMR 17952, the latter isolated from a rubber tire floating in seawater, were morphologically identified as *Amphichorda* sp. However, although they showed the typical morphological features of the genus (i.e., synnematous and mononematous conidiophores, flask-shaped conidiogenous cells with a strongly bent neck and solitary conidia that remain attached to the apex of the conidiogenous cell), they exhibited some morphological traits that did not exactly fit into any of the accepted species of *Amphichorda*.

### 3.1. Phylogeny

The molecular identification based on the BLAST search of our five unidentified strains revealed a high percentage of similarity with species of the genus *Amphichorda* using ITS sequences. Specifically, the percentage of identity was of 98% similarity to *A. cavernicola* (CGMCC 3.19571) and between 95 and 97% to other species of this genus. The molecular comparison using the LSU region revealed a 99% similarity to *O. coprophila* (CBS 247.82), *A. cavernicola* (CGMCC 3.19571) and between 98 and 99% similarity to other species of the genus *Amphichorda*. Other taxa closely related to our strains with a 97% identity with this locus are *Nigrosabulum globosum* (CBS 512.70), *Acremonium* (*Ac.*) *curvum* (GZUIFR 22.035) and *Ac. alternatum* (CBS 407.66). Members of the genera *Beauveria* and *Cordyceps* did not match with our sequences in the BLAST results. Despite this, we included them in the phylogenetic analysis due to the traditional placement of *Amphichorda* as a member of *Cordycipitaceae*. In addition, sequence analyses of these two gene markers allowed us to confirm or re-identify the CBS reference strains of *A. felina* as the following: CBS 110.08 and CBS 648.66 as *A. felina*, CBS 312.50 as *A. guana* and CBS 173.71 as *O. coprophila*.

Based on the BLAST results, we assessed the phylogenetic relationships among genera phylogenetically related to *Amphichorda* with the ITS and LSU regions. The resulting tree topologies from the individual analyses of these two gene markers were similar and did not show incongruences (Figures S1 and S2). Therefore, both alignments were concatenated into a single matrix. The final alignment of the concatenated ITS and LSU

regions comprised 65 taxa that included two representative strains from each *Amphichorda* species, as well as two representative strains from *O. coprophila* and the strains recovered from the marine environment, to prevent branch imbalance together with representative species belonging to the families *Bionectriaceae* and *Cordycipitaceae*. The tree was rooted with *Pochonia chlamydospora* (CBS 504.66) and *Metapochonia suchlasporia* (CBS 251.83) as outgroup. The total length comprised 1435 characters including gaps (ITS: 624, LSU: 811 characters). Among these, 906 characters were conserved sites (ITS: 255, LSU: 651), 529 characters were variable sites (ITS: 369, LSU: 160) and 417 characters were parsimony informative (ITS: 290, LSU: 127). For the ML analyses, the best-fit models were TIM2+F+I+G4 for the ITS region and TIM2e+I+G4 for the LSU region. For the BI analysis, the best-fit models were GTR+I+G for both the ITS and LSU region. Here, we represented the maximum-likelihood (RAxML) tree with the bootstrap support values of the ML analyses (RAxML and IQ-TREE) and Bayesian posterior probabilities at the nodes. The resulting phylogenetic tree resolved the genus *Amphichorda* as a monophyletic lineage within the family *Bionectriaceae* (Figure 1), being closely related to a well-supported clade that comprises two accepted genera in the family, *Nigrosabulum* and *Hapsidospora*, together with a recently described *Acremonium* species, *Ac. curvum* [42]. This latter species was, however, placed very distantly from the genus *Acremonium* s. str. The concatenated analysis defined five terminal clades within *Amphichorda*, where two marine strains (FMR 19404 and FMR 17952), representatives of our unidentified *Amphichorda* species, and those strains of *O. coprophila* (CBS 173.71 and CBS 247.82) represented two independent *Amphichorda* lineages. However, these molecular markers lacked the resolution to determine the phylogenetic relationships among *Amphichorda* species. Therefore, we performed a phylogenetic analysis, combining the ITS and LSU regions and the *tef*1 gene in order to delineate *Amphichorda* species with precision.

The individual ITS, LSU and *tef*1 alignments were concatenated into a single matrix because the resulting individual trees represented similar topologies (Figures S3–S5). The final ITS, LSU and *tef*1 alignment comprised the five unidentified *Amphichorda* strains, nine strains representative of the known *Amphichorda* species and three strains identified as *O. coprophila*. *Acremonium curvum* (GZUIFR 22.035), *Ac. globosisporium* (GZUIFR 22.036) and *Ac. sclerotigenum* (A101) were used as outgroup. The total length comprised 2168 characters including gaps (ITS: 506, LSU: 779, *tef*1: 883 characters). Among these, 1865 characters were conserved sites (ITS: 393, LSU: 727, *tef*1: 745), 303 characters were variable sites (ITS: 113, LSU: 52, *tef*1: 138) and 166 characters were parsimony informative (ITS: 55, LSU: 30, *tef*1: 81). For the ML analyses, the best-fit models were TNe+G4 for the ITS region, TNe+I for the LSU region and TN+F+G4 for the *tef*1 region. For the BI analysis, the best-fit models were K80+I for both the ITS and LSU regions and GTR+G for the *tef*1 region. Here, we represented the maximum-likelihood (RAxML) tree with the bootstrap support values of the ML analyses (RAxML and IQ-TREE) and Bayesian posterior probabilities at the nodes.

The resulting phylogenetic tree resolved the three species currently accepted in *Amphichorda* (*A. cavernicola*, *A. felina* and *A. guana*) as independent lineages (Figure 2). The five marine strains delineated an undescribed lineage within *Amphichorda* closely related to the clade representative of *O. coprophila*. The marine strains are proposed below as *Amphichorda littoralis*, and *O. coprophila* is accepted as an *Amphichorda* species. A detailed morphological characterization of the novel fungi is provided in the taxonomy section.

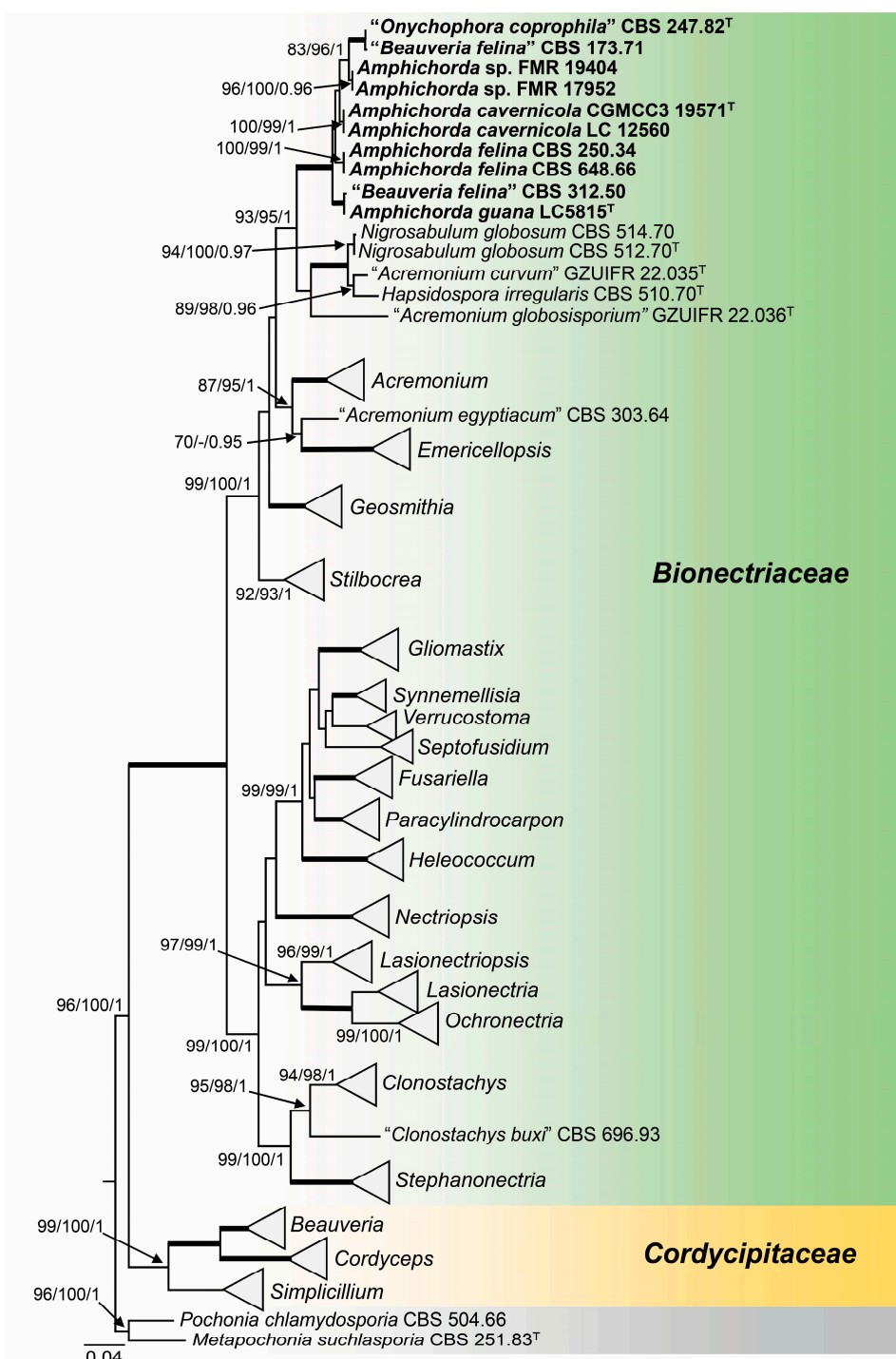

**Figure 1.** Phylogenetic tree inferred from a maximum-likelihood (RAxML) analysis based on a concatenated alignment of ITS and LSU sequences of 65 strains representing *Bionectriaceae*, *Cordycipitaceae* and outgroups. Numbers at the branches indicate support values (RAxML-BS/IQ-TREE-BS/BI-PP) above 70%/90%/0.95. The genus *Amphichorda* is printed in bold, and the other strains are collapsed based on their genera. Bold branches indicate full support values (100/100/1). [T] indicates ex-type strains. The tree is rooted to *Metapochonia suchlasporia* CBS 251.83 and *Pochonia chlamydosporia* CBS 504.66. Quote marks indicate strains with unresolved taxonomy. The scale bar represents the expected number of changes per site.

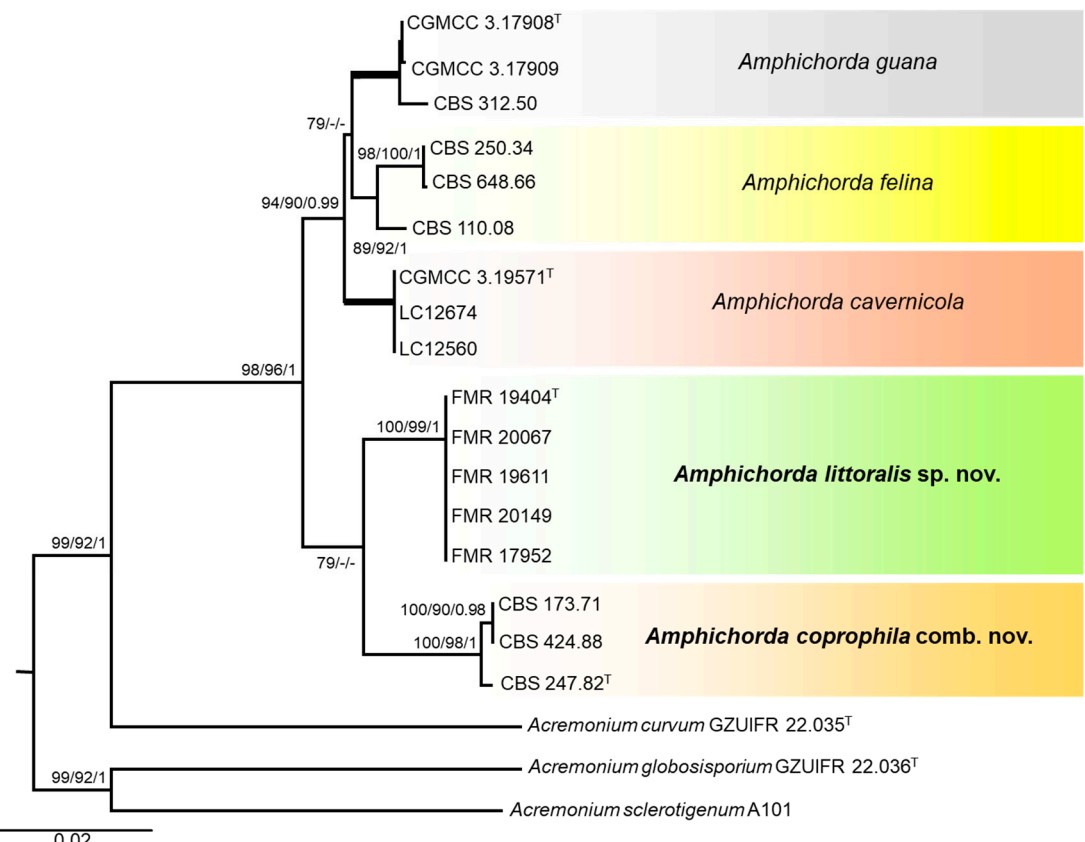

**Figure 2.** Phylogenetic tree inferred from a maximum-likelihood (RAxML) analysis based on a concatenated alignment of ITS, LSU and *tef*1 sequences of 20 strains representing *Amphichorda* and outgroups. Numbers at the branches indicate support values (RAxML-BS/IQ-TREE-BS/BI-PP) above 70%/90%/0.95. Bold branches indicate full support values (100/100/1). The novel species and combination are printed in bold. $^T$ indicates ex-type strains. The tree is rooted to *Acremonium curvum* (GZUIFR 22.035), *Acremonium globosisporium* GZUIFR 22.036 and *Acremonium sclerotigenum* A101. The scale bar represents the expected number of changes per site.

*3.2. Morphological Analysis*

In order to perform a morphological comparison of our strains, we reviewed the existing literature on *Amphichorda* and examined living cultures of the following species: *A. coprophila* (CBS 247.82, CBS 424.88 and CBS 173.71), *A. felina* (CBS 250.34, CBS 110.08 and CBS 648.66) and *A. guana* (CBS 312.50). Unfortunately, strains of *A. cavernicola* were not available for comparison. The colony color displayed across PDA, OA and SNA culture media represented the most accurate character to distinguish species. Microscopically, the reproductive structures were quite similar between species and consisted of flask-shaped conidiogenous cells that produce subglobose and smooth conidia. Although, there were subtle differences in the size range of these structures, they overlapped between species. In Table 2, we provide a synopsis of the key morphological characters that allow discrimination among species of *Amphichorda*.

**Table 2.** Synopsis of the morphological characters defining *Amphichorda* species.

| Species | Colony on PDA * | | Colony on OA/SNA * | Microscopic Features | | Citation |
|---|---|---|---|---|---|---|
| | Color | Diffusible Pigment | Color | Conidiogenous Cells Size (μm) | Conidia Size (μm) | |
| *A. cavernicola* | Cream yellow to sea shell | Not observed | White | 4.5–8 × 2–3 | 2.5–4 × 2–3.5 | [12]; this study |
| *A. coprophila* | Orange to brownish orange | Grayish orange | Light yellow | 6–10 × 2–2.5 | 3.5–5.5 × 2–3 | [15]; this study |
| *A. felina* | White | Grayish orange | White | 3–8.5 × 2–2.5 | 2.5–4.5 × 2–3.5 | [9]; this study |
| *A. guana* | White to yellowish | Yellowish | White | 7–10 × 2–3 | 4.5–5.5 × 3.5–5 | [11]; this study |
| *A. littoralis* | Greenish yellow | Light yellow | Greenish yellow | 5.5–11.5 × 1.5–2.5 | 2.5–4 × 2.5–3 | This study |

* Obverse side of the colony.

However, the pattern of conidiogenesis could not be properly determined under light microscopy. Therefore, we selected representative strains of *A. coprophila*, *A. felina*, *A. guana* and *A. littoralis* for examination under SEM (Figure 3).

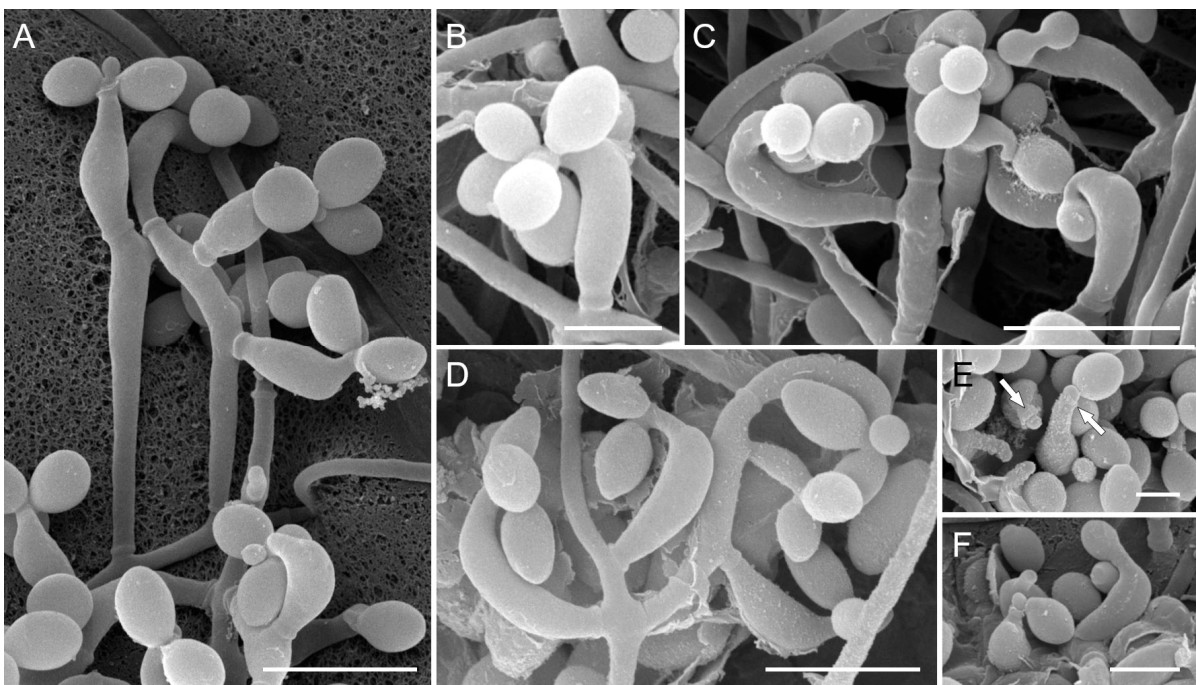

**Figure 3.** Scanning electron microscopy (SEM) of the reproductive structures from representative *Amphichorda* species. (**A**) Conidiophores, holoblastic conidiogenous cells and conidia in *Amphichorda felina* (CBS 250.34). (**B**) Holoblastic conidiogenous cell in *Amphichorda guana* (CBS 312.50). (**C**) Conidiophores, holoblastic conidiogenous cells and conidia in *Amphichorda littoralis* (FMR 20067). (**D**) Conidiophores, holoblastic conidiogenous cells and conidia from *Amphichorda coprophila* (ex-type CBS 247.82). (**E**,**F**) Enteroblastic (phialidic—arrows) and holoblastic conidiogenous cells in *Amphichorda coprophila* (CBS 424.88), respectively. The white arrows point at the enteroblastic collarettes. Scale bars: (**A**,**C**,**D**) = 5 μm; (**B**,**E**,**F**) = 2.5 μm.

Almost all the strains of *Amphichorda* species mentioned above showed a holoblastic pattern of conidiogenesis, producing few conidia in an apparent sympodial proliferation. However, the strains of *A. coprophila* differed in that the ex-type strain CBS 247.82 showed exclusively holoblastic conidiogenesis (Figure 3D), CBS 424.88 exhibited both holoblastic

(Figure 3F) and enteroblastic (phialidic) conidiogenous cells even showing a small collarette in the apex (Figure 3E). Another difference observed in these latter strains was the presence of either smooth or roughened conidiogenous cells. In the rest of the species examined, the conidiogenous cells were smooth.

### 3.3. Salt Tolerance Test

Considering the marine origin of *A. littoralis*, we compared its ability to grow under different NaCl concentrations with other species of the genus, predominantly isolated from terrestrial environments, to ascertain a possible preference for the marine environment. Although all *Amphichorda* strains managed to grow at up to 10% NaCl, each species showed different colony diameters across media. The colony diameter of *A. coprophila*, *A. felina* and *A. guana* decreased in an inverse proportion to the addition of NaCl. The only exception was the strain CBS 247.82 of *A. coprophila*, for which growth was restricted and quite similar across media with different salt concentrations. *Amphichorda littoralis* achieved similar maximum colony diameters across MEA and MEA supplemented with 3.5% and 5% NaCl. In particular, the strains FMR 19404, FMR 19611 and FMR 20067 reached the maximum colony diameter in MEA supplemented with 3.5% and 5% NaCl (Figure 4).

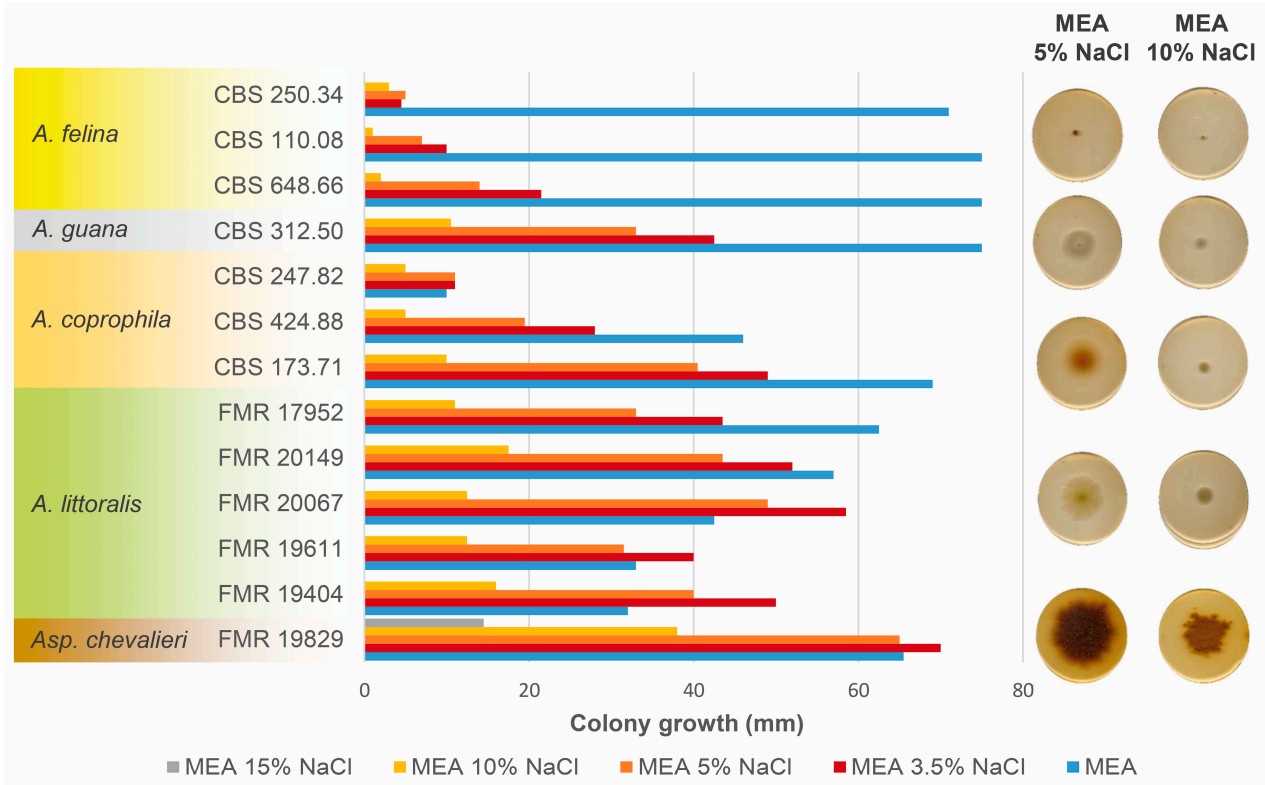

**Figure 4.** Mean colony diameter achieved by representative *Amphichorda* strains on malt extract agar (MEA) supplemented with different concentrations of NaCl after 4 weeks at 25 °C. The strain *Aspergillus chevalieri* (FMR 19829) was used as a positive control of growth in the different culture media. On the right side, culture plates on MEA5% and MEA10% are represented for representative strains from each species.

### 3.4. Taxonomy

**Amphichorda** Fries, Systema Orbis vegetalis 1:170 (1825)
=*Onychophora* W. Gams, P.J. Fisher & J. Webster, Transactions of the British Mycological Society 82 (1): 174 (1984)
*Type species*. *Amphichorda felina* (DC) Fries, Systema Orbis vegetalis 1: 170 (1825).
For synonyms of the species, see MycoBank database (https://www.mycobank.org/).
*Emended description*

*Asexual morph* with conidiophores synnematous or mononematous, semi-macronematous, erect, straight or flexuous, bearing lateral or terminal conidiogenous cells, arranged singly or in whorls, sometimes micronematous and reduced to conidiogenous cells growing directly from vegetative hyphae. *Conidiogenous cells* flask shaped, usually with a strongly bent neck, holoblastic, rarely enteroblastic, phialidic, hyaline, smooth walled or roughened. *Conidia* solitary, often remaining attached to the apex of the conidiogenous cell, subglobose, hyaline, smooth walled. *Sexual morph* not observed.

***Amphichorda coprophila*** (W. Gams, P.J. Fisher & J. Webster) Guerra-Mateo, Cano & Gené, comb. nov.

MycoBank: MB848789.

*Basionym*. *Onychophora coprophila* W. Gams, P.J. Fisher & J. Webster, Transactions of the British Mycological Society 82 (1): 174 (1984)

*Type*. ENGLAND, Devon, Dawlish Warren, from rabbit dung incubated at relative humidity of 95% for several weeks, Dec. 1981, *J. Webster* (holotype CBS H-1740 = IMI 275663, ex-type culture CBS 247.82).

*Asexual morph* on OA. *Mycelium* composed of smooth-walled, branched, septate, hyaline, 0.5–2(−3) μm wide hyphae. *Conidiophores* mononematous, arising directly from superficial mycelium, micronematous and reduced to conidiogenous cells growing directly or on a short lateral protrusion from vegetative hyphae or semi-macronematous, erect, straight or flexuous, commonly unbranched, bearing lateral or terminal conidiogenous cells, arranged singly or in whorls, hyaline and smooth walled. *Conidiogenous cells* flask shaped, usually with a strongly bent neck, holoblastic, 6–10 × 2–2.5 μm, hyaline, smooth walled; enteroblastic, phialidic, roughened under SEM, conidiogenous cells only observed in CBS 424.88. *Conidia* solitary, often remaining attached to the apex of the conidiogenous cell, subglobose to somewhat ellipsoidal, 3.5–4(−5.5) × 2–2.5(−3) μm, hyaline, smooth walled. *Sexual morph* not observed (adapted from Gams et al. [15]).

*Culture characteristics* (after 14 days at 25 °C). Colonies on PDA attaining 22–24 mm diam., slightly raised, irregularly sulcated, glabrous and brownish orange (7C5) at center (CBS 424.88 and CBS 173.71 orange (5B5)), velvety and white at periphery, margin crenate; reverse brownish orange at center and white at periphery; diffusible pigment grayish orange (6B5). On OA, colonies reaching 34–40 mm diam., flat, velvety, pale yellow (4A3) at center to white at periphery, margin entire and slightly lobated; reverse pale yellow. On SNA, colonies reaching 5–10 mm diam., glabrous, pale yellow, margin slightly lobated; reverse pale yellow.

*Additional specimens examined*. CANADA, Ontario, Landmark County, along Clyde River, from chipmunk dung, *K.A. Seifert* (CBS 424.88); ibid., Stoneleigh, from porcupine dung, Sep. 1969, *R.F. Cain* and *D.W. Malloch* (CBS 173.71).

*Notes*. *Amphichorda coprophila* is a well-supported species that represents a distant independent lineage in the genus *Amphichorda* (Figures 1 and 2). It can be morphologically distinguished by its orange to brownish-orange colonies on PDA (Table 2), the production of conidia through both holoblastic and phialidic conidiogenous cells (Figure 3D–F) and the occasional rough ornamentation of the conidiogenous cells under SEM. The original protologue reports the production of synnema-like tufts on OA [15]. However, we did not observe synnemata after 21 days in PDA, OA and SNA.

***Amphichorda littoralis*** Guerra-Mateo, Torres-Garcia, Cano & Gené, sp. nov. Figure 5.

MycoBank: MB 848035.

*Etymology*. Name refers to the area where this species was isolated, Mediterranean coast (Tarragona, Spain).

*Type*. SPAIN, Catalonia, Mediterranean coast, Tarragona, Platja del Miracle, N 41°6′19″, E 1°15′37″, from sediments at 20 m depth, Jun. 2021, *G. Quiroga-Jofre* and *D. Guerra-Mateo* (holotype CBS H-25254, ex-type culture FMR 19404, CBS 149935).

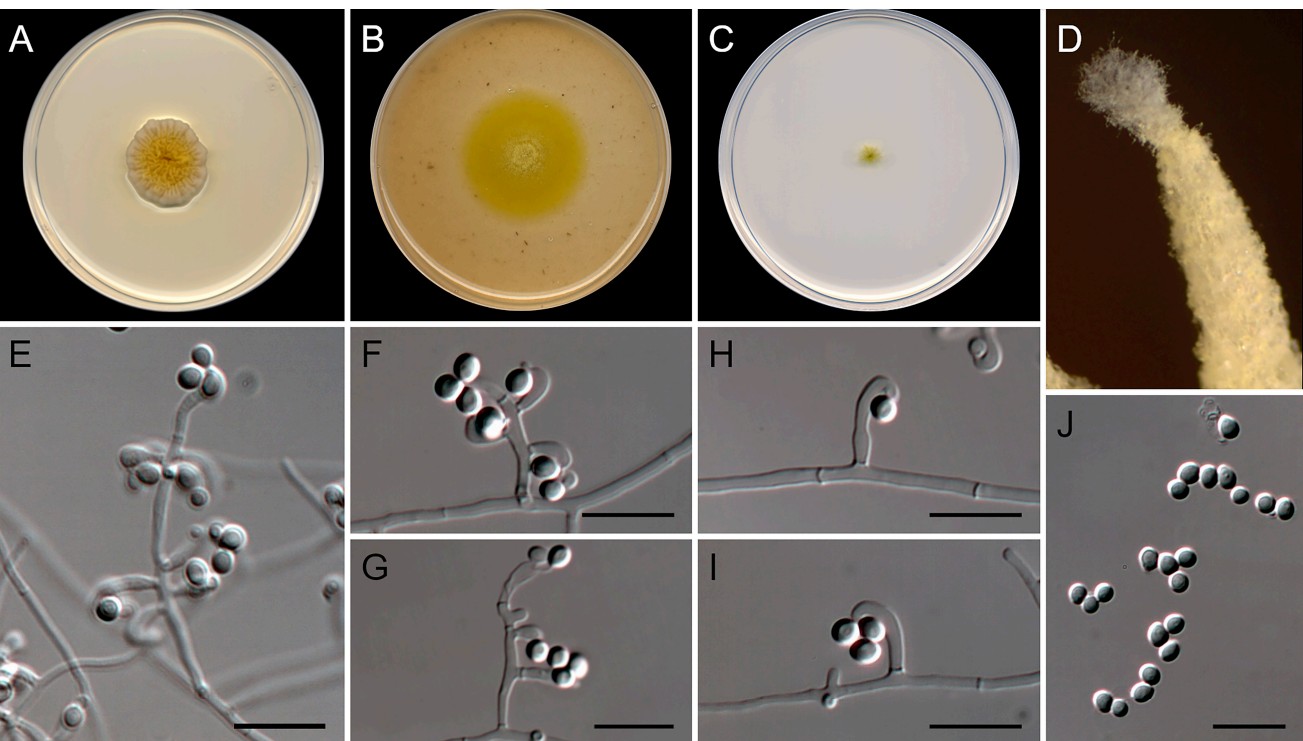

**Figure 5.** *Amphichorda littoralis* (ex-type FMR 19404). (**A**) Colony on PDA. (**B**) Colony on OA. (**C**) Colony on SNA after 14 d at 25 °C. (**D**) Synnema from strain FMR 20067. (**E**) Semi-macronematous conidiophore. (**F**,**G**) Micronematous conidiophores with attached conidia. (**H**,**I**) Conidiogenous cells growing directly from hyphae with attached conidia. (**J**) Conidia. Scale bars: 10 μm.

*Asexual morph* on OA. *Mycelium* composed of smooth-walled, branched, septate, hyaline, 1–1.5 μm wide hyphae. *Conidiophores* mononematous, rarely synnematous, arising directly from superficial mycelium, micronematous and reduced to conidiogenous cells growing directly or on a short lateral protrusion from vegetative hyphae or semi-macronematous, erect, straight or flexuous, commonly unbranched, bearing lateral or terminal conidiogenous cells, arranged singly or in whorls of 2–4, hyaline and smooth walled; synnematous conidiophores only observed in FMR 20067 on PDA at the margin of the colony, yellowish white, cylindrical with tomentose apex. *Conidiogenous cells* flask shaped, usually with a strongly bent neck, 6–10 (–11.5) × 1.5–2 μm, hyaline, smooth walled. *Conidia* solitary, often remaining attached to the apex of the conidiogenous cell, subglobose, 3–4 × 2.5–3 μm, hyaline, smooth walled. *Sexual morph* not observed.

*Culture characteristics* (after 14 days at 25 °C). Colonies on PDA attaining 20 mm diam., slightly raised, irregularly sulcated, glabrous and greenish yellow (1A7) at center, velvety (fasciculate in FMR 20067) and white at periphery, margin crenate; reverse greenish yellow (1A7) at center and white at periphery; diffusible pigment light yellow (4A4) produced after 21 days. On OA, colonies reaching 30–32 mm diam., flat, velvety, greenish yellow at center to grayish yellow at periphery, margin entire and slightly lobated; reverse greenish yellow (1A7). On SNA, colonies reaching 9–14 mm diam., glabrous, greenish yellow, margin slightly lobated; reverse greenish yellow (1A7).

*Additional specimens examined.* SPAIN, Catalonia, Mediterranean coast, Tarragona, Platja del Miracle, N 41°6′19″, E 1°15′37″, from sediments at 20 m depth, Oct. 2021, *G. Quiroga-Jofre* and *D. Guerra-Mateo* (FMR 19611); ibid., Platja de la Arrabassada, N 41°6′53″, E 1°16′48″, from sediments at 20 m depth, Jun. 2022, *G. Quiroga-Jofre* and *D. Guerra-Mateo* (FMR 20149); ibid., from sediments at 20 m depth, Jun. 2022, *G. Quiroga-Jofre* and *D. Guerra-Mateo* (FMR 20067); ibid., Mediterranean coast, Tarragona, from a fragment of floating rubber tire, Jul. 2020, *D. Torres-García* (FMR 17952).

*Notes*. *Amphichorda littoralis* is phylogenetically related to *A. coprophila* (Figures 1 and 2). Macroscopically, they can be distinguished by the color of the colony (Table 2). In the novel species, colonies are consistently greenish yellow across PDA, OA and SNA, while, in *A. coprophila*, colony color ranges from brown orange to pale yellow. Microscopically, the conidiogenous cells of *A. littoralis* are consistently smooth, while *A. coprophila* can show a rough ornamentation. Moreover, the phylogenetic distance between this novel species and other members of *Amphichorda* is around 96% for the *tef* 1 region and between 90 and 96% for the *BenA* region (Table S2 in Supplementary Material).

## 4. Discussion

In previous morphological and phylogenetic studies, the taxonomic circumscription of the genus *Amphichorda* has been controversial, and its phylogenetic relationships with other taxa have remained obscure. The most recent multi-locus phylogenetic tree assessing the diversity within *Cordycipitaceae*, included the type strain of *A. guana*, analyzed its LSU and *tef* 1 sequences and resolved *Amphichorda* as a distant independent lineage sister to *Cordycipitaceae* [4]. Our phylogenetic tree combining the ITS and LSU regions determined the close phylogenetic relationships of *Amphichorda* with the genera *Hapsidospora* and *Nigrosabulum* (Figure 1), two accepted members of the family *Bionectriaceae*. These results allowed us to recognize *Amphichorda* as a member of this family, despite it being the only representative genus with members producing conidia holoblastically. In the most recent review of the family, 41 genera were accepted, composed exclusively of fungi showing phialidic conidiogenous cells [2]. However, the order *Hypocreales* comprises members with asexual morphs, producing both enteroblastic (phialidic) and holoblastic conidiogenous cells. Although most of the families accepted in the order, such as *Clavicipitaceae*, *Ijuhyaceae*, *Myrotheciomycetaceae*, *Nectriaceae*, *Niessliaceae*, *Ophiocordycipitaceae*, *Sarocladiaceae*, *Stachybotriaceae*, *Stromatonectriaceae*, *Tilachlidiaceae* and *Xanthonectriaceae*, only show phialidic conidiogenesis, other families such as *Calcarisporiaceae*, *Cordycipitaceae* and *Hypocreaceae* show both types of conidiogenesis [1,2,4,43–47]. Only genera such as *Beauveria* and *Calcarisporium* exhibit holoblastic conidiogenous cells [2,48]. Thus, it seems that phialides represent the ancestral way of asexual reproduction in *Hypocreales*. In this sense, blastic conidiogenesis would have appeared independently as a secondary trait across different families. In particular, *Amphichorda* seems to represent a transition between both types of conidiogenesis for the *Bionectriaceae*. The type species of the genus, *A. felina*, and the rest of the species accepted show holoblastic conidiogenesis. The exception is *A. coprophila*, which can produce both types of condiogenous cells depending on the strain studied but, in particular, in the strain CBS 424.88. Here, we propose this species as a novel combination of the genus *Amphichorda*. However, as mentioned before, the original authors of the species already suggested the possible relation of this species with *Acremonium* (*Bionectriaceae*) [15]. They observed the conidiogenous cells of the species and concluded a phialidic conidiogenous pattern. Based on the close phylogenetic relationship of *A. coprophila* with holoblastic species, we can conclude with confidence that it can produce both types of conidiogenous cells. This trait, although odd, has already been described in *Bionectriaceae*, with the holoblastic mesoconidia described in some *Fusarium* species [49,50].

*Amphichorda* represents a group of morphologically cryptic species. The microscopic reproductive structures show subtle variations in their size range (Table 2). Thus, the best morphological character to distinguish species is the colony color across different culture media. We found this trait to be consistent across several strains on PDA, OA and SNA. However, colony color may be of little use in culture media such as MEA (Figure 4). For this reason, phylogenetic analyses represent the most accurate way to identify *Amphichorda* species. In particular, the ITS region is able to distinguish species of *Amphichorda* with precision, but other structural genes such as *tef* 1 and *BenA* can be used as secondary markers with similar results.

Correct phylogenetic identifications of species require DNA sequences obtained from type strains. This is a limitation when working with fungi described before the development

of DNA sequencing techniques. In our particular case, the type material of *A. felina* seems to be lost. We therefore selected representative strains identified as *A. felina* from the CBS culture collection, including some of coprophilous origin such as the protologue of this species, in order to determine a suitable candidate for epitypification and study the morphological and genetic variability of the species. However, the former goal was not feasible because the strains of coprophilous origin represented *A. guana* and *A. coprophila* (Table 1), and the strains that phylogenetically matched with *A. felina* did not correspond with the origin of the protologue, preventing the epitypification [14]. Despite this, the strain CBS 250.34 fits the morphological description of *A. felina*, and it has been extensively used to characterize the species in phylogenetic analyses. Therefore, we accept the strain CBS 250.34 as reference to stabilize the nomenclature of *A. felina* and, consequently, the genus *Amphichorda*, but its representation as type strain should be avoided.

Finally, we propose the novel species *A. littoralis*, the first species of the genus de-scribed from the marine environment. Marine fungi are described as those that can grow and sporulate under marine-like conditions [51]. All the strains that represent this species have been isolated from the marine habitat, and all strains managed to grow and sporulate at up to MEA 10% NaCl. We understand this as indicative of a preference for the marine environment and consider this species to possibly represent a marine fungus. This species updates the ecological range of the genus. *Amphichorda* was previously recognized as a group of parasitic and coprophilous fungi. Although a pathogenic behavior has been described for *A. felina* [52], we conclude that it is predominantly composed of saprotrophic fungi with a preference for substrates with abundant organic matter such as dung or marine sediments.

## 5. Concluding Remarks

Our phylogenetic analyses combining the ITS and LSU regions revealed the genus *Amphichorda* as a member of the family *Bionectriaceae*, where it represents the only holoblastic group. The combination of morphological and phylogenetic analyses determined our marine strains as a novel species, *A. littoralis*, and resolved the taxonomic position of *O. coprophila* as a new member of the genus *Amphichorda*. The current study is the largest sampling of *Amphichorda* ever subjected to multi-locus sequence analyses, provides a comprehensive phylogenetic backbone and represents a framework for future studies on the genus.

**Supplementary Materials:** The following supporting information can be downloaded at: https://www.mdpi.com/article/10.3390/d15070795/s1. Table S1: GenBank accessions of representative taxa from *Bionectriaceae*, *Cordycipitaceae* and outgroups included in the phylogenetic analyses. Table S2: Phylogenetic distance between *Amphichorda* species for the *BenA* region. Figure S1: Phylogenetic tree inferred from a maximum-likelihood (RAxML) analysis based on an alignment of ITS of 63 strains representing *Bionectriaceae*, *Cordycipitaceae* and outgroups. Figure S2: Phylogenetic tree inferred from a maximum-likelihood (RAxML) analysis based on an alignment of LSU of 60 strains representing *Bionectriaceae*, *Cordycipitaceae* and outgroups. Figure S3: Phylogenetic tree inferred from a maximum-likelihood (RAxML) analysis based on an alignment of ITS sequences of 20 strains representing *Amphichorda* and outgroups. Figure S4: Phylogenetic tree inferred from a maximum-likelihood (RAxML) analysis based on an alignment of LSU sequences of 20 strains representing *Amphichorda* and outgroups. Figure S5: Phylogenetic tree inferred from a maximum-likelihood (RAxML) analysis based on an alignment of *tef* 1 sequences of 20 strains representing *Amphichorda* and outgroups [53–73].

**Author Contributions:** Conceptualization, D.G.-M., J.F.C.-L. and J.G.; methodology, D.G.-M., J.G. and J.F.C.-L.; software, D.G.-M. and J.F.C.-L.; validation, J.F.C.-L. and J.G.; formal analysis, D.G.-M., J.F.C.-L. and J.G.; investigation, D.G.-M., J.F.C.-L. and J.G.; resources, V.B., J.F.C.-L. and J.G.; data curation, D.G.-M., J.F.C.-L. and J.G.; writing original draft preparation, D.G.-M. and J.G.; writing review and editing, D.G.-M., J.F.C.-L. and J.G.; visualization, J.F.C.-L. and J.G.; supervision, J.F.C.-L. and J.G.; project administration, V.B. and J.G.; funding acquisition, J.G. All authors have read and agreed to the published version of the manuscript.

**Funding:** This study was supported by the grant PID2021-128068NB-100 funded by MCIN/AEI/10, 13039/501100011033/ and by "ERDF A way of making Europe".

**Institutional Review Board Statement:** Not applicable.

**Data Availability Statement:** Not applicable.

**Acknowledgments:** The authors thank the CBS culture collection (The Netherlands) and its curators for providing some fungal strains included in the study and to Gabriel Quiroga-Jofre for services in the collection of the samples.

**Conflicts of Interest:** The authors declare no conflict of interest.

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
