# Peer review of "Phylogeny and Taxonomy of the Genus Amphichorda (Bionectriaceae): An Update on Beauveria-like Strains and Description of a Novel Species from Marine Sediments"

_diversity, doi:10.3390/d15070795_

Round 1

Reviewer 1 Report

The manuscript solved the taxonomy of the genus Amphichorda well based on their comprehensive sequence data, locating the genus in the family Bionectriaceae. A novel species and a new combination are proposed in Amphichorda. Their study also revised the morphological characteristics of the genus, expending it to cover both holoblastic and enteroblastic conidiogenous cells.

A few other comments:

1) Since Amphichorda coprophila has been modified, a complete description will be helpful for others to recognise this species.  

2) Before this study, the morphology of Amphichorda is consistent between the species in the genus. Therefore, the sentence “Despite this, morphological differences between Amphichorda species are confusing……” in the introduction is confusing. Please delete it.

Author Response

The manuscript solved the taxonomy of the genus Amphichorda well based on their comprehensive sequence data, locating the genus in the family Bionectriaceae. A novel species and a new combination are proposed in Amphichorda. Their study also revised the morphological characteristics of the genus, expending it to cover both holoblastic and enteroblastic conidiogenous cells.

Response: Thank you very much for the comments on our study and suggestions to improve the manuscript.

A few other comments:

1) Since Amphichorda coprophila has been modified, a complete description will be helpful for others to recognise this species.  

Response: A description of Amphichorda coprophila has been added to the text, including the variations on the protologue according to our observations.

2) Before this study, the morphology of Amphichorda is consistent between the species in the genus. Therefore, the sentence “Despite this, morphological differences between Amphichorda species are confusing……” in the introduction is confusing. Please delete it.

Response: The sentence has been deleted.

Reviewer 2 Report

Tha paper entitled "Phylogeny and taxonomy of the genus Amphichorda (Bionectriaceae): an update on beauveria-like strains and description of a 3 novel species from marine sediments" represents a novelty, it describes a novel genus and 3 new species. It is generally well witten and presented. 

The major issue concerns the Intrduction that I feel is too long and with details that could go to the discussion section.

Minor comments are in the attached PDF.

For these reasons I recommend minor revisions

Dear Editor,

The paper entitled "Phylogeny and taxonomy of the genus Amphichorda (Bionectriaceae): an update on beauveria-like strains and description of a 3 novel species from marine sediments" represents a novelty, it describes a novel genus and 3 new species. It is generally well witten and presented. 

The major issue concerns the Intrduction that I feel is too long and with details that could go to the discussion section.

Minor comments are in the attached PDF.

For these reasons I recommend minor revisions

Author Response

The paper entitled "Phylogeny and taxonomy of the genus Amphichorda (Bionectriaceae): an update on beauveria-like strains and description of a 3 novel species from marine sediments" represents a novelty, it describes a novel genus and 3 new species. It is generally well witten and presented.

Response: Thank you very much for the comments and suggestions to improve the content of our manuscript.

The major issue concerns the Intrduction that I feel is too long and with details that could go to the discussion section.

Response: We appreciate your suggestion to move part of the introduction to the discussion section. However, if we transfer the information suggested, the reader may find it difficult to understand part of our study. Therefore, if the reviewer does not mind, we would prefer to leave the taxonomic history of Amphichorda and the information about Onychophora coprophila in the introduction in order to make the content of the Results more comprehensible for the readers. However, if the reviewer and the editor do not agree with our explanation, we can adapt the text accordingly.

Minor comments are in the attached PDF. For these reasons I recommend minor revisions

Response: All the comments have been introduced in the text.

Reviewer 3 Report

I thank the authors and the editor for the opportunity to review this manuscript. I have really enjoyed reading it and I think it is a nice piece of paper, with a well-executed taxonomic exercise.

The manuscript deals with the problem in the circumscription of various taxa of the Cordycipitaceae family; and tries to gather morphological, phylogenetic and structural criteria in a corpus that pays tribute to the resolution of the problem of the species. The authors try to circumscribe coherently the genus Amphicorda -and some problematic neighbors- using a polyphasic approach, MLST, structure and morphology as their estimators.

The strengths of the manuscript are: Provide an updated taxonomic corpus for the genus Amphichorda, describe a new species-specific context, and correct the classification of Onychophora coprophila through a new combination. Importantly, Amphichorda is repositioned in to Bionectriaceae family.

I really liked the handling of the taxonomic hypothesis throughout the manuscript; the taxonomic markers used for the phylogenetic analysis are scientifically sound and except for a few inconsequential comments from me, I believe it has the quality to be published by Diversity (ISSN 1424-2818) with minor modifications.

Minor comments:

1.       In the abstract, line 25: “… and propose the new combination Amphichorda coprophila”. It should say: …and propose Amphichorda coprophila com. nov.

2.       Lines 207-208: I think a very brief clarification should be made about the practical use of similarity values with the ITS or rRNA (18S, 28S) to assign specie-specific categories. Although the authors mention that it was used as a preliminary identification criterion, this can confuse the amateur reader, and lead to the belief that “similarity level of ≥ 99% with ≥ 90% of sequence cover” could indeed, be a universal rule for species-level identification. As we well know, such similarity values are not resolving enough to delimitate species in fungi.

3.       By the way, the new accessions reported in the table 1 (OQn), they not appear on NCBI so far (14/06/2023), so I couldn't verify the blast similarity relationships.

4.       Line 221:  It is a clean exercise to visually check the concordance between different "gene trees" when working with individual markers. A less biased way for this, would be execute “statistical speciation tests” under different models; such as the Bayesian Poisson Tree Processes or GMYC. Are the phylogenetic hypotheses obtained by the authors consistent post speciation test?

6.       Could the authors share their crude phylogenetic reconstructions in newick format (as they did with their alignments)? This is only with the intention of subjecting the trees to speciation tests aforementioned. Special interest arises the strains CBS 110.08 A. feline and CBS 247.82 A. coprophila, and the last one in particular for its unique phenotype in conidiation.

7.       Figure 3, the mentioned arrows are not black.

8.       I do not fully agree with the general statement that ITS is the "fungal barcode"; if this were universal - and true - there would not be so much taxonomic chaos in the group. I also believe that the authors are only seeing the effect of a clade (Amphichorda) still little studied and little represented at the level of specimens and sequences; therefore, the saturation and low resolution that is frequently seen with the ITS in other more famous fungal groups (Trichoderma, Aspergillus...) has not yet manifested in Amphichorda. That will change when the number of descriptions increases.

Final comment:

I repeat that I consider this manuscript an excellent work. But I would like to conclude with a comment about the concept of species and the approach to the problem of the species -which is itself the problem of the manuscript-. Rather than mentioning a functional species concept for fungi (we know that the problem is very complex...), I do believe necessary among mycologists to evaluate other hypotheses that bring us closer to canonical and known speciation models, in an effort to get closer to an integrative taxonomy (Chethana, K.W.T., Manawasinghe, I.S., Hurdeal, V.G. et al. What are fungal species and how to delineate them?. Fungal Diversity 109, 1–25 (2021). https://doi.org/10.1007/s13225-021-00483-9). The polyphasic pathway has been useful for many years and we owe a lot, but it is necessary to enter the genomic era in fungal taxonomy. Sequencing a fungal genome today is quite affordable (~200 USD in America by ONT). A genome, even a draft, has information not for a few markers -like SSU, IST, LSU, tef1 or BenA- but hundreds of markers that better represent the evolutionary history of any taxon. From the genome, population characteristics can be explored, such as allelic frequency, gene flow, recombination. I consider that categories vital on the phenomenon of speciation; Importantly, we can evaluate the genomic coherence between contexts, especially in sympatric species, where the asexual morphs of fungi have a large niche. Amphichorda is a clade with few representatives(yet). It may be worth sequencing their genomes and evaluating the phylogenetic hypothesis from phylogenomics; this, together with the phenotypic and structural criteria, should consolidate the taxonomy of the group.

Author Response

The manuscript deals with the problem in the circumscription of various taxa of the Cordycipitaceae family; and tries to gather morphological, phylogenetic and structural criteria in a corpus that pays tribute to the resolution of the problem of the species. The authors try to circumscribe coherently the genus Amphicorda -and some problematic neighbors- using a polyphasic approach, MLST, structure and morphology as their estimators.

The strengths of the manuscript are: Provide an updated taxonomic corpus for the genus Amphichorda, describe a new species-specific context, and correct the classification of Onychophora coprophila through a new combination. Importantly, Amphichorda is repositioned in to Bionectriaceae family.

I really liked the handling of the taxonomic hypothesis throughout the manuscript; the taxonomic markers used for the phylogenetic analysis are scientifically sound and except for a few inconsequential comments from me, I believe it has the quality to be published by Diversity (ISSN 1424-2818) with minor modifications.

Response: Thank you very much for your kind words, your specific comments on our study and your suggestions to improve the manuscript.

Minor comments:

  1. In the abstract, line 25: “… and propose the new combination Amphichorda coprophila”. It should say: …and propose Amphichorda coprophila nov.

Response: This sentence of the abstract has been updated like you suggested.

  1. Lines 207-208: I think a very brief clarification should be made about the practical use of similarity values with the ITS or rRNA (18S, 28S) to assign specie-specific categories. Although the authors mention that it was used as a preliminary identification criterion, this can confuse the amateur reader, and lead to the belief that “similarity level of ≥ 99% with ≥ 90% of sequence cover” could indeed, be a universal rule for species-level identification. As we well know, such similarity values are not resolving enough to delimitate species in fungi.

Response: The sentence has been modified to specify that the levels of similarity and sequence cover used in the study are specific for Amphichorda species.

  1. By the way, the new accessions reported in the table 1 (OQn), they not appear on NCBI so far (14/06/2023), so I couldn't verify the blast similarity relationships.

Response: The sequences will not be available for public use until the publication of the article or the privacy policy of GenBank with our sequences expires. Despite this, we provide the sequences as an attached file for their revision.

  1. Line 221:  It is a clean exercise to visually check the concordance between different "gene trees" when working with individual markers. A less biased way for this, would be execute “statistical speciation tests” under different models; such as the Bayesian Poisson Tree Processes or GMYC. Are the phylogenetic hypotheses obtained by the authors consistent post speciation test?

Response: Thank you for the suggestion to use speciation tests to analyze our data. We tried the Bayesian Poisson Tree Processes with 500 000 MCMC generations, 0.25 burn-in, 123456 seed value and checked that the MCMC chains converged. We tested the individual ITS, LSU and TEF Amphichorda trees with and without outgroups and obtained unconclusive results without posterior probability support. However, when we tried to plot the resulting concatenated ITS-LSU-TEF tree without outgroups, the program determined with the ML tree A. cavernicola, A. guana, A. coprophila and A. littoralis as independent species with the exception of A. felina, where the strain CBS 110.08 was determined as an independent species. Although, this result was not supported by posterior probabilities. Based on this result and the data in our manuscript, we understand that A. felina may represent a complex of species. However, we think that with the current information about the genus Amphichorda, it is safer to consider the strain CBS 110.08 as A. felina.

  1. Could the authors share their crude phylogenetic reconstructions in newick format (as they did with their alignments)? This is only with the intention of subjecting the trees to speciation tests aforementioned. Special interest arises the strains CBS 110.08 A. feline and CBS 247.82 A. coprophila, and the last one in particular for its unique phenotype in conidiation.

Response: We provide the crude phylogenetic reconstructions in Newick format with and without outgroups in the attached file.

  1. Figure 3, the mentioned arrows are not black.

Response: Figure 3 caption has been updated to include the color of the arrows displayed in the figure.

  1. I do not fully agree with the general statement that ITS is the "fungal barcode"; if this were universal - and true - there would not be so much taxonomic chaos in the group. I also believe that the authors are only seeing the effect of a clade (Amphichorda) still little studied and little represented at the level of specimens and sequences; therefore, the saturation and low resolution that is frequently seen with the ITS in other more famous fungal groups (Trichoderma, Aspergillus...) has not yet manifested in Amphichorda. That will change when the number of descriptions increases.

Response: We agree that the term “fungal barcode” may be misleading, for this reason we have replaced “ITS barcode” for “ITS region” throughout the text as suggested the reviewer.

Final comment:

I repeat that I consider this manuscript an excellent work. But I would like to conclude with a comment about the concept of species and the approach to the problem of the species -which is itself the problem of the manuscript-. Rather than mentioning a functional species concept for fungi (we know that the problem is very complex...), I do believe necessary among mycologists to evaluate other hypotheses that bring us closer to canonical and known speciation models, in an effort to get closer to an integrative taxonomy (Chethana, K.W.T., Manawasinghe, I.S., Hurdeal, V.G. et al. What are fungal species and how to delineate them?. Fungal Diversity 109, 1–25 (2021). https://doi.org/10.1007/s13225-021-00483-9). The polyphasic pathway has been useful for many years and we owe a lot, but it is necessary to enter the genomic era in fungal taxonomy. Sequencing a fungal genome today is quite affordable (~200 USD in America by ONT). A genome, even a draft, has information not for a few markers -like SSU, IST, LSU, tef1 or BenA- but hundreds of markers that better represent the evolutionary history of any taxon. From the genome, population characteristics can be explored, such as allelic frequency, gene flow, recombination. I consider that categories vital on the phenomenon of speciation; Importantly, we can evaluate the genomic coherence between contexts, especially in sympatric species, where the asexual morphs of fungi have a large niche. Amphichorda is a clade with few representatives(yet). It may be worth sequencing their genomes and evaluating the phylogenetic hypothesis from phylogenomics; this, together with the phenotypic and structural criteria, should consolidate the taxonomy of the group.

Response: We truly appreciate your comment and the reference included in it. We agree that phylogenomics represents a powerful tool to assess biodiversity from several complementary points of view. We are looking forward to delving into the amazing world of phylogenomics and hope to be able to publish taxonomic studies in this sense in the future.